# Structural and Heat Transfer Model Analysis of Wall-Mounted Solar Chimney Inlets and Outlets in Single-Story Buildings

**Zhicheng Fang, Wanjiang Wang \*, Yanhui Chen and Junkang Song**

School of Architecture and Engineering, Xinjiang University, Urumqi 830047, China
\* Correspondence: wangwanjiang@xju.edu.cn

**Abstract:** Wall-mounted solar chimneys use solar radiation to heat the air inside the chimney cavity and use thermal pressure to create natural convection. Applying this principle allows for the indoor ventilation of a building without energy consumption. However, in wall-mounted solar chimney designs, different air inlet and outlet design dimensions can have varying degrees of impact on the effectiveness of wall-mounted solar chimney ventilation. In order to analyze the internal airflow state and airflow temperature field distribution of wall-mounted solar chimneys, physical models of wall-mounted solar chimneys with six different air outlet-to-inlet cross-sectional area ratios were developed in this research work. Before numerical simulation analysis, heat transfer analysis of the wall-mounted solar chimney's structural components and airflow channels was carried out, and corresponding mathematical heat transfer models were established. The internal flow state and temperature distribution characteristics of a wall-mounted solar chimney were analyzed by steady-state simulations using the computational fluid dynamics software, Ansys Fluent. Finally, transient simulation calculation analysis was conducted under six different S-value models to investigate the variation in the natural ventilation of a single-story building's wall-mounted solar chimney for a whole day. The study showed that under the same simulation conditions, $80\% \leq S < 100\%$ effectively avoided the formation of vortices in the internal airflow of the wall-mounted solar chimneys and kept the ventilation effect of wall-mounted solar chimneys at a high level. The results of this study provide a reference for the optimization of research on the design of the air inlet and outlet structures of wall-mounted solar chimneys for single-story buildings.

**Keywords:** solar chimneys; natural ventilation; CFD; energy-efficient buildings



## 1. Introduction

With the intensification of energy consumption, environmental problems are also gradually being highlighted [1]. Global warming and the frequent emergence of extreme weather all remind us to save on energy consumption, and the rational use of clean energy has become a trend of the times [2,3]. In China, energy consumption from building operations accounts for 20% of total energy consumption [4]. In order to reconcile energy consumption with development, it is urgent to achieve energy efficiency in buildings [5]. The application of natural ventilation technology in buildings is a breakthrough in developing green and low-carbon buildings [6]. Natural ventilation mainly makes the airflow inside a building through thermal or wind pressure, ensuring the supply of fresh air inside the building without consuming any energy and producing any pollution [7]. This also makes natural ventilation technology the primary choice for improving the indoor thermal environment, reducing indoor temperature and humidity, and improving indoor air quality [8]. The traditional means of natural ventilation mainly include controlling the opening of the doors and windows of buildings to encourage the outdoor air of buildings to enter rooms; however, the effectiveness of this natural ventilation method is greatly restricted by people's lifestyle in practical application [9]. A solar chimney is an excellent solution to this problem. A solar chimney connects the heating cavity to the indoor space. Solar radiation

is used to heat the air in the cavity, which makes the air buoyant under thermal pressure, thus driving the airflow into the room [10]. As an effective passive natural ventilation technology, the research on solar chimneys has great significance in creating energy-saving buildings [11].

The two main types of solar chimney applications in buildings are wall-mounted solar chimneys and roof-mounted solar ones [12,13]. According to recent research reports, the main research objective of solar chimneys is usually to analyze and improve the natural ventilation effect of solar chimneys. Most studies on the operational effectiveness of solar chimneys have investigated the effect of external influences on the ventilation effect of solar chimneys by controlling the energy input. These major external influences include environmental factors, solar radiation, and building structure [14,15]. Zamora investigated the effect of solar chimneys in determining wind speed and direction by numerical methods, determined that the effect of wind is roughly related to wind speed, and gave a numerical model for practical engineering applications [16]. Alex investigated the effect of ambient wind speed on the performance of solar chimneys and showed that ambient wind speed only affects the air velocity of solar chimneys at greater than 2.00 m/s [17]. Solar radiation is an essential external influence factor, so it is crucial to establish the quantitative relationship between solar radiation and airflow rate in the practical design of engineering. Duan et al. developed a mathematical model to predict the relationship between airflow rate and solar radiation in solar chimneys [18]. Long et al. investigated the effect of room configuration on the effectiveness of natural ventilation in solar chimneys. It was shown that natural ventilation was most effective when windows and air intake were located in the middle of the wall [19].

The aforementioned literature focuses on the influence of external factors on the operational effectiveness of solar chimneys. However, the actual application of solar chimneys shows that the actual operational performance effect of solar chimneys is not only affected by external factors, but also, their dimensional structural design can restrict the improvement of the solar chimney ventilation effect [20,21]. As Khosravi suggested in his paper, the cavity width of a wall-mounted solar chimney, the ratio between the inlet and outlet areas, and the height of a wall-mounted solar chimney all affect the solar chimney ventilation performance [22]. Sivalakshmi et al. conducted an experimental analysis of indoor ventilation performance for different air passage widths in the chimney by comparing four air passage widths. It was shown that the maximum wind speed of the wall-mounted solar chimney was 0.62 m/s at a 15-cm air passage width, while the maximum air mass flow rate was 0.042 kg/s [23]. Hussain et al. investigated the operational performance of a wall-mounted solar chimney system with four different inlet configurations. The study's results showed that the air velocity and mass flow rate in the wall-mounted solar chimney were higher than for the other three inlet configurations; when the air inlet had a vertical cross-section, the system operated the best [24].

In summary, an overall analysis of current research reports on solar chimneys shows that no relevant studies have been conducted to analyze the design of the air inlet and outlet dimensions of wall-mounted solar chimneys for single-story buildings [25,26]. Therefore, to further investigate the effects of the air inlet and outlet dimensions of wall-mounted solar chimneys on the operational effectiveness of wall-mounted solar chimney systems, the main research objectives of this paper included the following five points:

(1) A wall-mounted solar chimney with an outer glass cover was designed, and the ratio of the inlet and outlet cross-sectional area of the wall-mounted solar chimney is S;

(2) Six physical models of wall-mounted solar chimneys with different S-values were developed based on the different ratios of the inlet and outlet cross-sectional areas of wall-mounted solar chimneys;

(3) The energy input and output processes of each component of the wall-mounted solar chimney were analyzed, and the analytical equations for the heat transfer of each component in the wall-mounted solar chimney were established;

(4)     Ansys Fluent steady-state simulations analyzed the air temperature distribution and flow conditions in the wall-mounted solar chimney at different S-values;

(5)     Finally, Ansys Fluent transient simulation was applied to analyze the natural ventilation effect of the wall-mounted solar chimney at each moment of actual operation under different S-values.

## 2. Model Description and Heat Transfer Analysis

### 2.1. Physical Model Description

The wall-mounted solar chimney with a glass cover consists of a 3-m high building envelope, an insulated shell, a plexiglass cover made of methacrylate with 92% transmittance, and an air inlet and outlet, as shown in Figure 1.

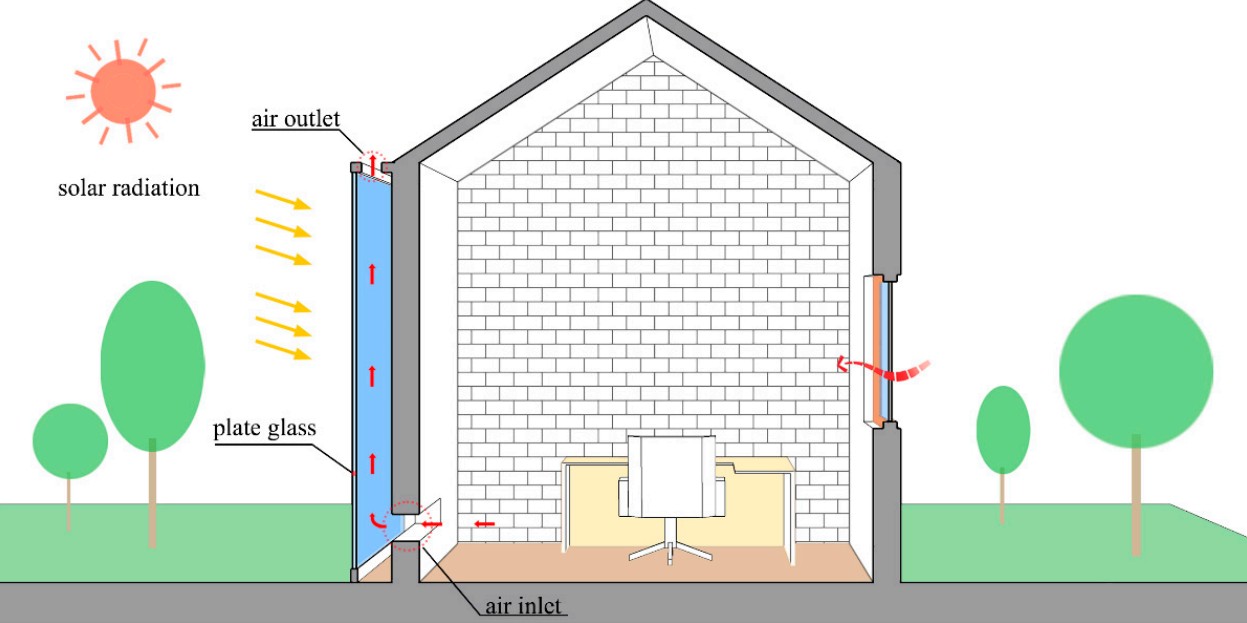

**Figure 1.** Diagram of the wall-mounted solar chimney.

The air inside the building enters the internal heating cavity from the air inlet at the bottom of the wall-mounted solar chimney. After being heated by solar radiation, the air inside the cavity forms an upward buoyancy force and flows from the wall-mounted solar chimney air outlet to the outside environment; this enables the building to be naturally ventilated without energy consumption [14,27].

In order to better grasp the setting of boundary conditions in the subsequent numerical simulation, it is necessary to establish a dynamic heat transfer mathematical model of each component by analyzing the heat transfer process of each component of the wall-mounted solar chimney. As shown in Figure 2, there are three types of heat transfer equilibrium equations in the overall structure of the wall-mounted solar chimney. They are the glass cover heat balance equation, the wall heat balance equation, and the cavity area heat balance equation.

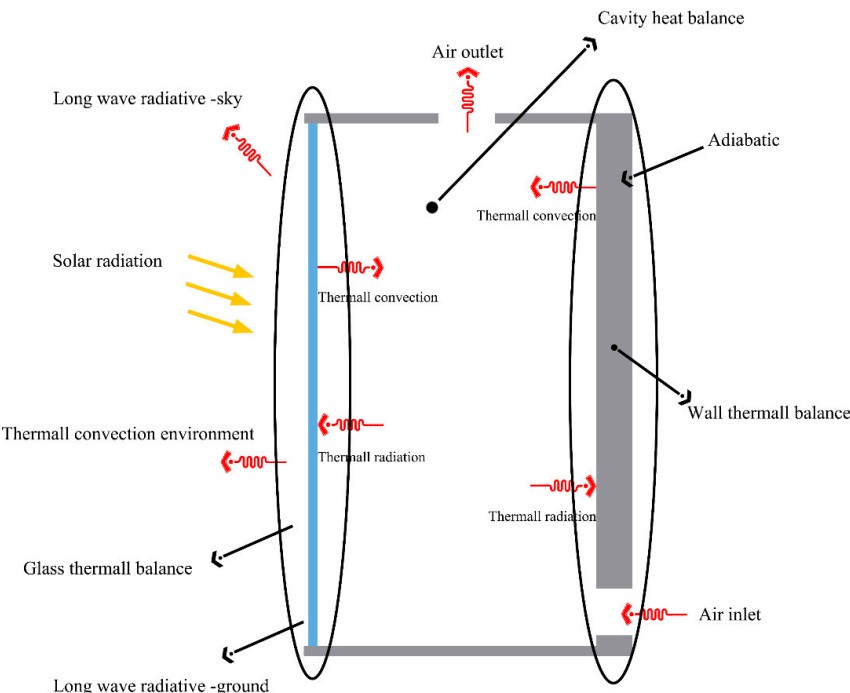

**Figure 2.** Wall-mounted solar chimney heat transfer analysis diagram.

### 2.2. Thermal Balance Analysis of Glass Cover

In the energy input balance of the glass cover, the energy input mainly comes from the solar radiation energy absorbed by the glass cover and the radiation heat exchange from the wall; the energy output includes the convection heat exchange between the glass cover and the inner air, and the convection heat exchange between the outer and inner ambient air [28]. Longwave radiation heat exchange between the glass cover, and the sky and ground, is also the primary way of energy output from the glass cover [8,29]. As a result, the heat balance equation for the glass cover was established as shown in Equation (1):

$$IA_s\alpha_1 + Q_{rad.wall.2} = Q_{out.air.1} + Q_{in.air.1} + Q_{sky.1} + Q_{groud.1} \tag{1}$$

In the above formula, $I$ is the amount of solar radiation, $A_s$ is the area of the glass cover, $\alpha_1$ is the absorption rate of the glass cover, $Q_{rad.wall.2}$ is the radiant heat transfer with both the glass cover and wall, $Q_{out.air.1}$ is the convective heat exchange from the exterior of the glass cover, $Q_{in.air.1}$ is the convection of heat within the glass cover, $Q_{sky.1}$ is the longwave amount of radiation of both the glass cover and the sky, and $Q_{groud.1}$ is the longwave radiation between the glass cover and the ground [28,30–32].

$$Q_{out.air.1} = h_{out.1}A_s(T_b - T_a) \tag{2}$$

$$Q_{in.air.1} = h_{in.1}A_s(T_b - T_m) \tag{3}$$

$$Q_{sky.1} = h_{1.sky}A_s(T_b - T_{sky}) \tag{4}$$

$$Q_{groud.1} = h_{1.groud}A_s(T_b - T_{ground}) \tag{5}$$

In the above formulas, $T_b$ is the average temperature of the glass cover, $T_a$ is the outdoor air temperature of the building, and $T_{ground}$ is the outdoor ground temperature of the building. $T_{sky}$ is the temperature of the sky and $T_m$ is the average temperature of the air inside the chimney. $h_{out.1}$ is the coefficient of heat transfer by the convection of both the glass cover and the air outside the building, and $h_{in.1}$ is the coefficient of heat convection between the chimney's glass cover and the air in the flow channel within the chimney [33]. $h_{1.sky}$ is the exchange of radiation that takes place for both the exterior of the glass cover and the sky, and $h_{1.groud}$ is the coefficient of heat radiation transfer of both the Earth's

surface and the outside of the glass cover [34]. Each heat transfer coefficient was calculated according to the following formula:

$$h_{out.1} = 5.7 + 3.8 U_\infty \tag{6}$$

$$h_{1.sky} = \varepsilon_1 \sigma_b (T_b + T_{sky})(T_b{}^2 + T_{sky}{}^2) \tag{7}$$

$$h_{1.groud} = \varepsilon_1 \sigma_b (T_b + T_{ground})(T_b{}^2 + T^2{}_{ground}) \tag{8}$$

If Re ≤ 2300:

$$N_u = 4.9 + \frac{0.0606(\text{RePr}D_h/H)^{1.2}}{1 + 0.0909(\text{RePr}D_h/H)^{0.7}\text{Pr}^{0.17}} \tag{9}$$

If Re > 2300:

$$N_u = 0.0158\text{Re}^{0.8} \tag{10}$$

$U_\infty$ is the outdoor airflow rate, $\varepsilon_1$ is the emissivity of the glass cover surface, and $\sigma_b$ is the Stefan–Boltzmann constant. $N_u$ is the air convection heat exchange strength between the glass cover and the cavity area, Re is the Reynolds number, and Pr is the Prandtl number. $D_h$ is the characteristic length of the air basin inside the chimney, and $H$ is the contact height between the glass cover and the air basin.

### 2.3. Wall Heat Balance Analysis

In the wall's energy balance, the energy input comes from the solar radiation energy transmitted through the glass cover. The energy output includes the convective heat exchange that occurs in the air and the wall in the flow channel located within the chimney, as well as the radiation exchange between the wall and the inner surface of the glass cover's heat [33,35]. This formula expresses the specific heat balance equation:

$$IA_s\alpha_2\beta_1 = Q_{rad.wall.2} + Q_{in.air.2} \tag{11}$$

$$Q_{in.air.2} = h_{in.2}A_s(T_u - T_m) \tag{12}$$

$\alpha_2$ is the rate of the wall's absorption, $\beta_1$ is the glass cover's transmittance, $Q_{in.air.2}$ is the transmission of heat by convection that takes place between the wall and the air in the flow channel, $T_m$ is the average temperature of the air that flows through the chimney flow channel, $T_u$ is the average temperature of the wall surface, and $h_{in.2}$ is the wall-to-air convective heat transmission coefficient [33,36].

### 2.4. Heat Balance Analysis of Airflow Region

When establishing the heat balance equation for the airflow channel area, we assumed that the airflow in the flow channel was an incompressible continuous medium flow, and the temperature difference between the fluid and air in different areas inside the solar chimney was significant [37,38]. Therefore, the energy input in the airflow channel was the convection heat exchange with the inner side of the glass cover and the wall, and the temperature difference between the air formed the heat exchange process.

$$Q_{out.air.1} + Q_{in.air.2} + Q_w = mc_p(T_m - T_{in}) + mc_p(T_m - T_a) \tag{13}$$

$$Q_w = \frac{R_{in.1}}{R_{in.1} + R_{out.2}}(I\alpha_1 + I_s\alpha_2) \tag{14}$$

$Q_w$ is the solar chimney's air absorbs heat, $I_s$ is the solar scattered radiation illumination, $T_{in}$ is the mean interior air temperature, and $T_a$ is the average outside building air temperature [39,40]. $m$ represents the mass flow of the air, $c_p$ is the air-specific heat capacity under constant pressure, $R_{in.1}$ is the internal barrier to heat transmission on the glass cover, and $R_{out.2}$ is the heat transfer thermal resistance of the outer surface of the glass cover [19,24,41].

## 3. Simulation Method

In this numerical simulation study, the airflow and heat transfer inside the wall-mounted solar chimney cavity were simulated by Ansys Fluent computational fluid dynamics software [42]. Finally, the variation in the airflow field and temperature distribution inside the wall-mounted solar chimney was analyzed by combining the cloud display and the simulation data results [43]. The air outlet cross-sectional area to air inlet cross-sectional area was defined as S. Six models with different S-values of 20%, 40%, 60%, 80%, 100%, and 120% were created in the model processing software, SpaceClaim, for subsequent simulations.

### 3.1. Simulation Condition Setting

Due to the complex airflow and heat transfer process in the solar wall chimney, adding some subtle simulation conditions had little effect on the final result of the simulation, but did increase the number of grids and the difficulty of the calculation convergence. Therefore, in order to simplify the simulation calculation process, it was necessary to make the following reasonable assumptions:

(1)	The air inside the wall-type solar chimney is an ideal incompressible gas, usually using the Boussinesq assumption;
(2)	Ignore the viscous dissipation of air in the flow channel;
(3)	The physical properties of the air inside the solar wall chimney are constant, and the air density is the only variable;
(4)	Ignore the shell's heat transfer loss and the outer wall's radiant heat loss.

### 3.2. Physical Computation Grid Validation

In this simulation study, SpaceClaim software was applied to establish the physical model of the wall-mounted solar chimney, and the unstructured polyhedral form was used for meshing in Fluent Mesh. As shown in Figure 3, the air inlet and outlet of the physical model of the solar wall chimney were subjected to boundary layer encryption processing; the orthogonal mesh quality of the six models all exceeded 0.90; and therefore, the mesh quality met the computational requirements of the simulation study.

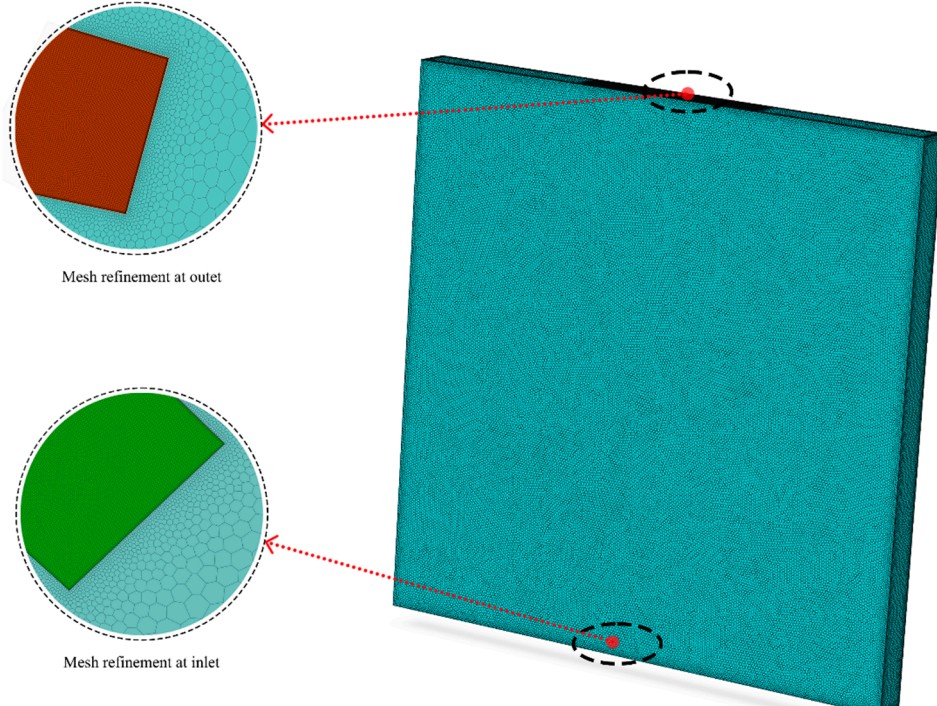

**Figure 3.** The meshing of the physical model.

In order to ensure the accuracy of the simulation results and further explore the influence of the grid size on the discrete error of the calculation results, it was necessary to verify the grid independence. The number of grids verified under different models is shown in Table 1. Five sets of grids with different numbers were generated under each of the six physical models, and the natural ventilation calculated by different grids was used as the result for comparison. The verification results are as follows in Figure 4.

**Table 1.** The number of grids verified under different models.

| Model | Number of Grids |
|---|---|
| S = 20% | N1 = 1,500,325  N2 = 2,912,532<br>N3 = 3,598,568  N4 = 3,766,896  N5 = 4,885,214 |
| S = 40% | N1 = 1,476,323  N2 = 2,596,512<br>N3 = 3,432,521  N4 = 3,710,120  N5 = 4,832,314 |
| S = 60% | N1 = 1,442,361  N2 = 2,493,256<br>N3 = 3,364,896  N4 = 3,642,321  N5 = 4,765,425 |
| S = 80% | N1 = 1,411,532  N2 = 2,437,452<br>N3 = 3,201,458  N4 = 4,064,521  N5 = 4,654,232 |
| S = 100% | N1 = 1,386,526  N2 = 2,013,652<br>N3 = 3,186,754  N4 = 3,341,263  N5 = 4,412,589 |
| S = 120% | N1 = 1,346,897  N2 = 2,298,546<br>N3 = 2,965,236  N4 = 3,515,478  N5 = 4,146,582 |

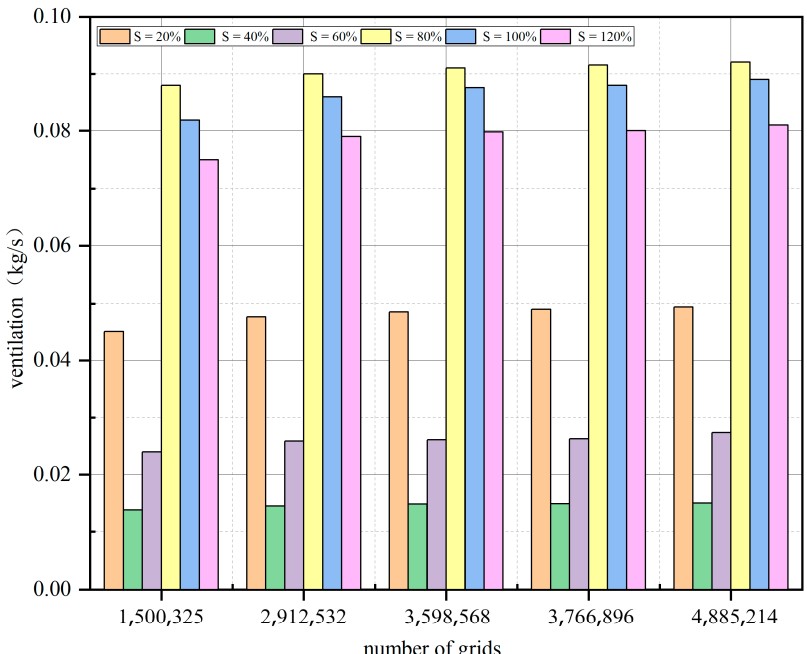

**Figure 4.** Calculated ventilation under different numbers of grids.

Through the above grid-independent study, in Figure 5, it can clearly be seen that the errors of the third and fourth sets of grids for S = 20%, S = 40%, S = 80%, S = 100%, and S = 120% were 0.92%, 0.74%, 0.55%, 0.51%, and 0.33%, respectively. The error between the second set and the third set of the grid calculation results at S = 60% was 0.78%.

A comprehensive analysis of the above calculations was performed, in order to obtain the most accurate simulation results with limited computational resources. The six models, S = 20%, S = 40%, S = 60%, S = 80%, S = 100%, and S = 120%, were selected to simulate with the corresponding grids of N4 = 3,766,896, N3 = 3,432,521, N3 = 3,364,896, N4 = 4,064,521, N4 = 3,341,263, and N4 = 3,515,478, respectively.

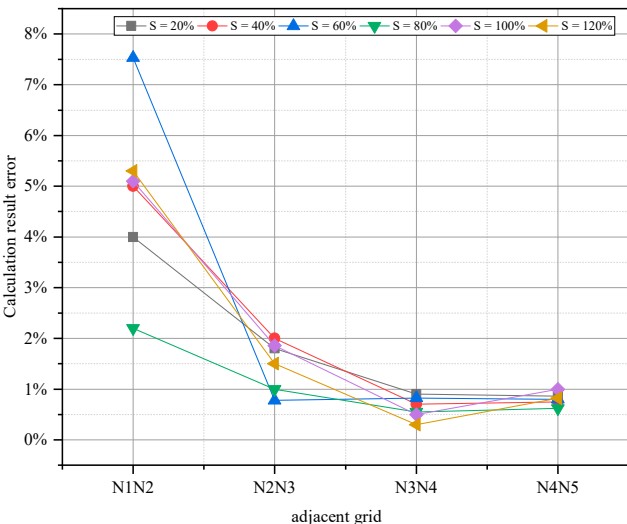

**Figure 5.** The computational error of adjacent grids.

### 3.3. Governing Equations

Inside a wall-mounted solar chimney, the flow of air is governed by three main principles: the conservation of mass equation, the conservation of momentum equation, and the conservation of energy equation [44,45], as shown in Equations (15)–(18). The air in the cavity of the wall-mounted solar chimney appeared to flow irregularly in the direction of the mainstream due to the different degrees of temperature difference at each location, forming turbulence, at which time the air fluid inside the wall-type solar chimney also needed to follow the turbulence transport control equations shown in Equations (19) and (20).

(1) Conservation of mass equation

$$\frac{\partial u_x}{\partial x} + \frac{\partial u_y}{\partial y} + \frac{\partial u_z}{\partial z} = 0 \tag{15}$$

(2) Conservation of momentum equation

$$\frac{\partial(\rho u_x)}{\partial t} + \frac{\partial(\rho u_x^2)}{\partial x} + \frac{\partial(\rho u_x u_y)}{\partial y} + \frac{\partial(\rho u_x u_z)}{\partial z} = -\frac{\partial p}{\partial y} + \mu\left(\frac{\partial^2 u_y}{\partial x^2} + \frac{\partial^2 u_y}{\partial y^2} + \frac{\partial^2 u_y}{\partial z^2}\right) + \rho_0 g - \rho\beta(T - T_0)g \tag{16}$$

$$\frac{\partial(\rho u_z)}{\partial t} + \frac{\partial(\rho u_x u_z)}{\partial x} + \frac{\partial(\rho u_y u_z)}{\partial y} + \frac{\partial(\rho u_z^2)}{\partial z} = -\frac{\partial p}{\partial z} + \mu\left(\frac{\partial^2 u_z}{\partial x^2} + \frac{\partial^2 u_z}{\partial y^2} + \frac{\partial^2 u_z}{\partial z^2}\right) \tag{17}$$

$u_x$, $u_y$, $u_z$ is the velocity of the fluid in the x, y, z direction and $\mu$ is the fluid dynamic viscosity coefficient.

(3) Energy and heat transfer equation

$$\frac{\partial(\rho E)}{\partial t} + \nabla\cdot(\vec{v}\cdot(\rho E + p)) = \nabla(\eta_f \nabla T - \sum_n h_n \vec{N} + (\bar{\bar{\tau}}_f \cdot \vec{v})) + S_h \tag{18}$$

$\rho$ is the air density, $p$ is fluid pressure, $E$ is the fluid energy, $\eta_f$ is the actual value of thermal conductivity, $\vec{N}$ is the diffusive flux of component $N$, and $S_h$ is the heat that was produced by the source of volume heat [40,46,47].

(4) Turbulent kinetic energy control equation

$$\frac{\partial(\rho k)}{\partial t} + \frac{\partial(\rho k u_i)}{\partial x_i} = \frac{\partial}{\partial x_i}\left(\alpha_k \mu_{eff}\frac{\partial k}{\partial x_i}\right) + G_k - \rho\varepsilon \tag{19}$$

$$\frac{\partial(\rho\varepsilon)}{\partial t} + \frac{\partial(\rho\varepsilon u_i)}{\partial x_i} = \frac{\partial}{\partial x_i}\left(\alpha_\varepsilon \mu_{eff}\frac{\partial\varepsilon}{\partial x_i}\right) + \frac{C_{1\varepsilon}}{k}G_k - C_{2\varepsilon}\rho\frac{\varepsilon^2}{k} \tag{20}$$

$k$ is the turbulent pulsation kinetic energy; $\varepsilon$ is the dissipation rate of turbulent pulsating momentum; $\mu_{eff}$ is the effective dynamic viscosity of the fluid; $\alpha_\varepsilon$ is the reciprocal of the effective Planter number of $\varepsilon$; $\alpha_k$ is the reciprocal of the $k$ effective Planchet number; and $G_k$ is the kinetic energy of the turbulent flow that is produced by the average velocity gradient. $C_{1\varepsilon}$ and $C_{2\varepsilon}$ were the default constants for calculation, taking 1.32 and 1.41, respectively, and $\alpha_\varepsilon$ was the rough Prandtl number of the dissipation rate, taking 1.2 [48,49].

### 3.4. Solver and Boundary Condition Settings

The numerical simulation study was carried out using Ansys Fluent 2021 R1 software [50]. Considering gravity based on a pressure basis, the turbulence model was selected as Realizable K-Epsilon SST, the radiation model was selected as the solar radiation model with DO discrete coordinates, and the SIMPLE pressure–velocity coupling algorithm was used to solve the discretized control equations [51–53]. In steady-state analysis, the solar calculator in the solar radiation model can be used to calculate the solar irradiance in different periods and regions; in this simulation study, the city of Urumqi, China, which is located at 86°37′33″–88°58′24″ E and 42°45′32″–44°08′00″ N, was used as the simulation study site. The solar irradiance of Urumqi at 12:00 on 15 June is shown in Table 2.

**Table 2.** Solar irradiance in Urumqi at 12:00 on 15 June 2022.

| Direction | Direct Irradiation | Vertical Scattered Radiation | Horizontal Scattered Radiation | Ground Vertical Reflected Radiation |
|---|---|---|---|---|
| Solar irradiance (W/m$^2$) | 853.125 | 100.462 | 112.172 | 81.6977 |

The boundary condition settings were as follows:

(1) The air intake utilized the pressure intake border, the pressure at the inlet was 0 Pa, the inlet airflow temperature was 30 °C, and the inlet size was 100 mm × 1000 mm;

(2) The air outlet was the pressure outlet so in order to study the natural ventilation effect caused by the hot pressure, the outlet pressure was set to 0 Pa, and the outlet size was set to six different sizes according to the S-value;

(3) The size of the glass cover was 3000 mm × 3000 mm × 5 mm, the heat boundary was the convective-radiation mixed boundary, and the convection coefficient was calculated and determined according to the above formula, while the radiation boundary was the semi-transparent radiation boundary, and the inner and outer emissivity were both 0.55;

(4) The thickness of the wall body was 130 mm, the boundary conditions of radiation convection were mixed, and the above formula determined the convective heat transfer coefficient;

(5) The surrounding protective structure was the boundary condition of four adiabatic walls with equal thicknesses, which were 5 mm.

## 4. Results and Discussion

### 4.1. Overall Comparison of S-Values—Temperature Field Distribution

The temperature field is a comprehensive reflection of the heat transfer and heat absorption of the fluid air inside a solar chimney wall. Under the six groups of physical models with different S-values, the simulated changes in the temperature distribution of the air inside the solar chimney are shown in Figure 6.

When the S-value was less than 80%, the air temperature inside the wall-mounted solar chimney gradually increased along the vertical wall direction. Due to the buoyancy force, the highest air temperature inside the wall-mounted solar chimney reached the maximum at the exit end. Combined with Figure 7, we found that when S = 20%, the wall-mounted

solar chimney air outlet temperature was 38.05 °C, compared to at S = 80%, when air outlet temperature was 1.8% higher, which was due to the sudden increase in pressure at the outlet caused by the local temperature rise. At S = 20%, the maximum local heat flow density at the top of the wall-mounted solar chimney cavity was 215.775 W/m$^2$. As the value of S gradually increased, when S = 60%, the maximum local heat flow density in the internal cavity of the wall-mounted solar chimney was only 169.353 W/m$^2$, which was 27.4% lower than the maximum heat flow density in the internal cavity of the wall-mounted solar chimney at S = 20%. At this time, the local temperature rise in the wall-mounted solar chimney was slight, and the difference in the buoyancy of the internal airflow was not apparent.

Further increasing the S-value, the local heat flow density of the internal cavity of the wall-mounted solar chimney reached 187.951 W/m$^2$ at S = 80%, which was 10.9% higher than the maximum heat flow density of the internal cavity of the wall-mounted solar chimney at S = 60%. At this time, the internal cavity of the wall-mounted solar chimney had a significantly higher local air temperature in the internal cavity of the chimney due to the sharp increase in local heat flow density. At S = 100% and S = 120%, the maximum local heat flow density of the internal cavity of the wall-mounted solar chimney was as high as 235.106 W/m$^2$ and 239.105 W/m$^2$, with an increased ratio of more than 25.2% and 27.2%, respectively. At the same time, the local air temperature in the wall-mounted solar chimney cavity increased rapidly, making a vast temperature difference between the inlet air temperature and the original air inside the chimney, and the buoyancy force increased with the increase in temperature difference. The pressure at the outlet of the wall-mounted solar chimney air climbed, causing a secondary reflux of air in the internal cavity of the chimney.

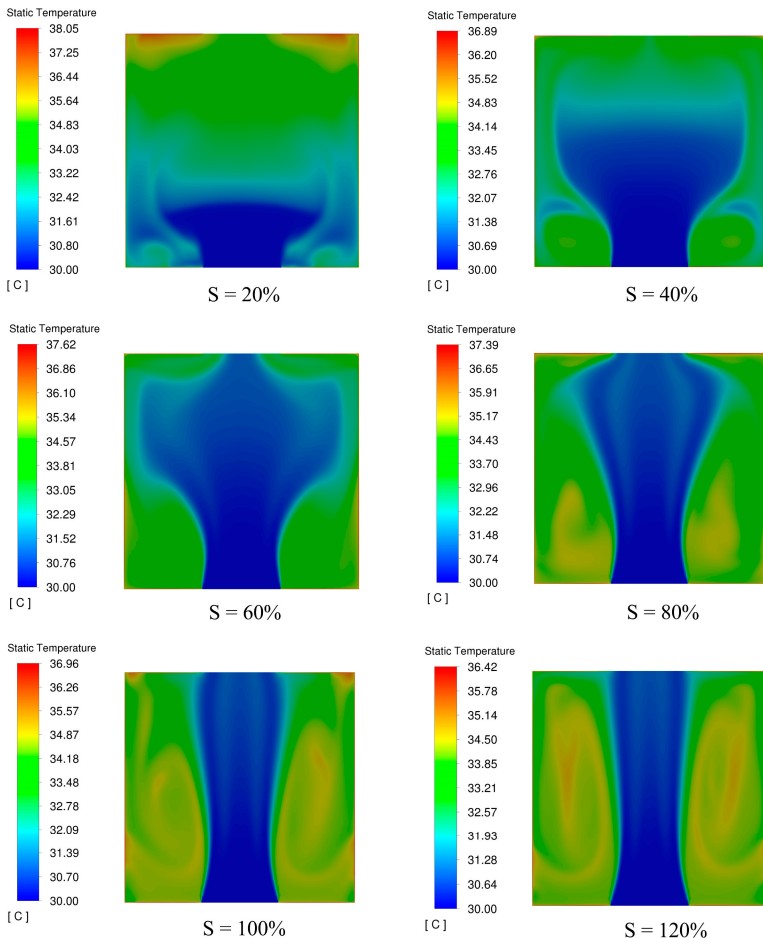

**Figure 6.** Distribution maps of the internal temperature field of the wall-mounted solar chimney under different S-values.

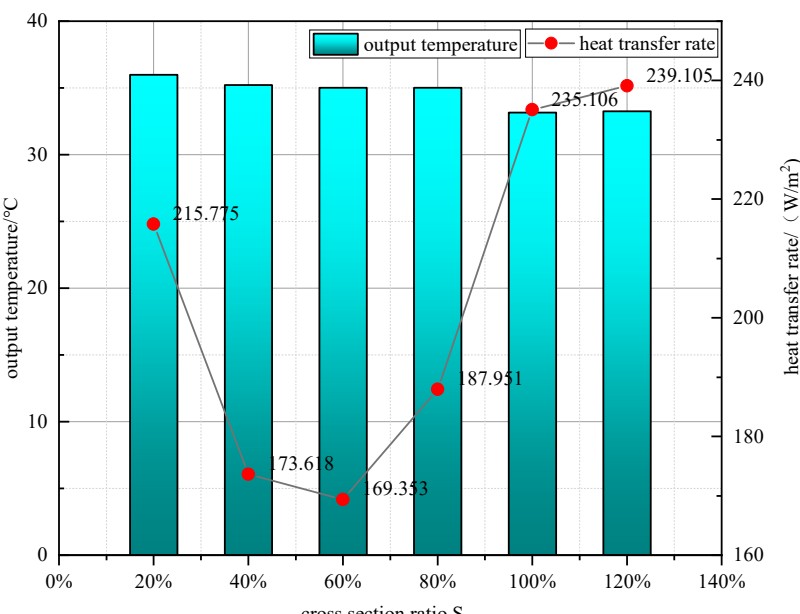

**Figure 7.** Variation in the outlet temperature and heat flux of wall-mounted solar chimney under different S-values.

As S > 80% of the air inside the wall-mounted solar chimney at the outlet produced a sharp increase in secondary reflux, the air inside the cavity in the main flow channel on both sides exhibited local warming and the phenomenon of airflow stagnation. Wall-mounted solar chimney cavity airflow in the formation of a vortex for wall-mounted solar chimney natural ventilation caused a significant loss of efficiency. In order to avoid the formation of vortices in the airflow inside the wall-mounted solar chimney, the phenomenon of localized warming of the air inside the wall-mounted solar chimney must be changed. This air vortex phenomenon can be well regulated by controlling the S-value, between 80% and 100% of the air vortex inside the wall-mounted solar chimney, while keeping the air buoyancy inside the chimney at a high level.

*4.2. Overall Comparison of S-Values—Flow Field Analysis*

In natural convection, the velocity field is usually coupled with the associated temperature field since the volume force term in the equation contains the temperature variable associated with it when solving the momentum differential equation. Simulation of the velocity field can visualize the velocity and direction of airflow inside a wall-mounted solar chimney, which is essential for studying the natural ventilation inside a wall-mounted solar chimney. The variation of the simulated airflow distribution inside the solar chimney under six different sets of S-value physical models is shown in Figure 8.

By comparing the numerical simulation results under different S-values, it was analyzed that when S < 80%, the airflow direction inside the wall-mounted solar chimney cavity flowed uniformly from the inlet to both sides of the primary flow channel air. The air flowed along the wall toward the outlet under the action of the buoyancy force, and the airflow velocity was similar everywhere inside the wall-mounted solar chimney at this time. When S > 80% of the wall-mounted solar chimney inside the mainstream channel airflow, velocity increased rapidly, and the airflow vortex was formed on both sides near the mainstream channel.

In Figure 9, as the value of S increased, its inlet velocity also became larger and larger, while the outlet velocity became smaller and smaller; the maximum outlet velocity was obtained at S = 20%, of 1.89611 m/s, while the minimum outlet velocity was obtained at S = 120%, of 1.5833 m/s—a decrease of 16.5% year-by-year. When S = 80%, the maximum value of wall-mounted solar chimney ventilation was obtained in the simulation study as 0.091 kg/s.

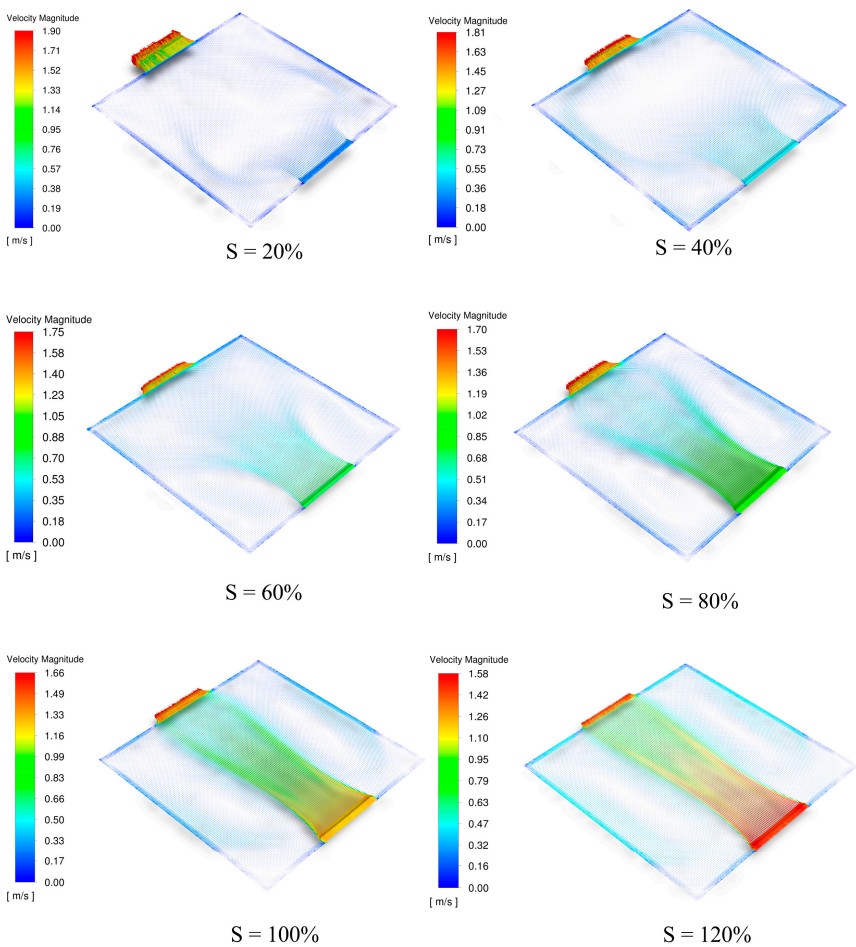

**Figure 8.** The internal velocity field distribution maps of the wall-type solar chimney under different S-values.

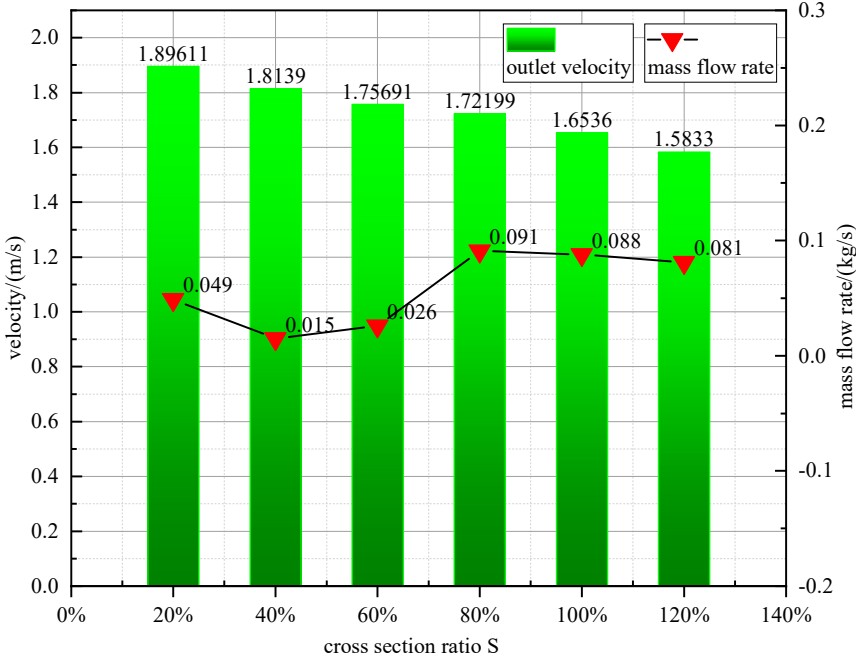

**Figure 9.** Variation in air velocity and air mass flow rate at the outlet of a wall-mounted solar chimney for different S-values.

To further analyze the effect of the S-value on the average air mass flow rate inside a wall-mounted solar chimney the least squares principle was used to combine the average air mass flow rate of the wall-mounted solar chimney for the six models mentioned above with the S-values of the corresponding models. A polynomial fit was performed using MatLab mathematical analysis software. The results of the fitted curves are shown in Figure 10. By comparing the fitting accuracy $R^2$ under the three polynomial fits, the fitting accuracy $R^2$ of the fitted curves reached 0.98 when the scattered parameters were fitted using a degree-five polynomial. The results were analyzed comprehensively, and the polynomial was selected to establish the mathematical model expressions for the average air mass flow rate inside the wall-mounted solar chimney for different S-values.

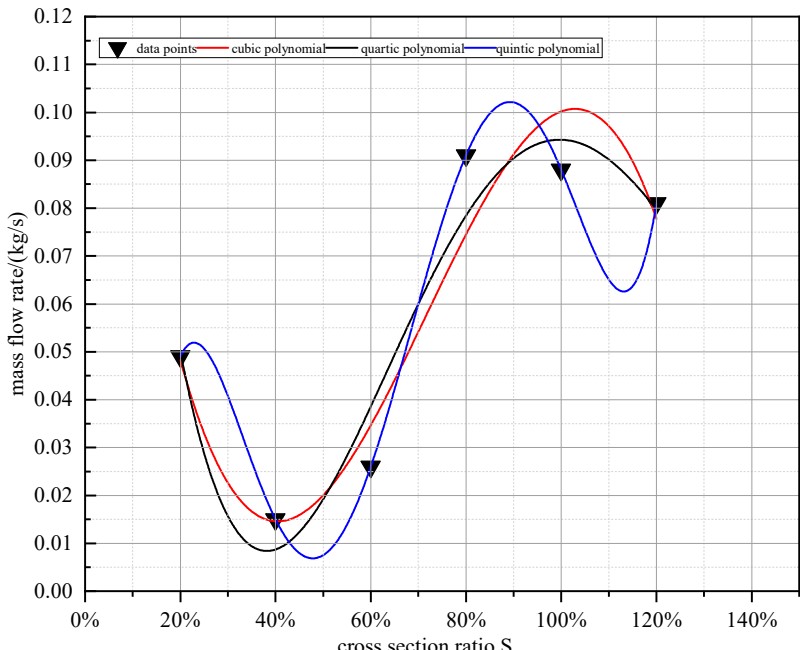

**Figure 10.** The fitting curve of the S-values and air mass flow rates.

Based on the quintuple polynomial fit results, the mathematical relationship between the average air mass flow rate inside the wall-mounted solar chimney and the value of S can be expressed as:

$$M = 8.25521s^5 - 28.17708s^4 + 35.0781s^3 - 19.1979s^2 + 4.5586s - 0.329 \tag{21}$$

$M$ denotes the magnitude of the air mass flow rate extracted from the building interior by the wall-mounted solar chimney. $s$ denotes the wall-mounted solar chimney outlet-to-inlet cross-sectional area ratio.

### 4.3. Analysis of Actual Ventilation Rate

In order to derive the magnitude of natural ventilation for each moment of operation of the six different S-value models in the actual building, further transient simulation analysis was made based on the above steady-state simulation. The numerical simulation boundary conditions were still adopted as the above steady-state boundary conditions, and the simulation location was still selected as Urumqi, Xinjiang, China. The simulation time was kept the same as the steady-state simulation, which was selected from 07:30 to 20:30 on 15 June 2022. The initial ambient air temperature was 28 °C and the maximum solar radiation was 1144.5 W/m$^2$ [54]. The air mass flow rate in the wall-mounted solar chimney at specific moments is shown in Figure 11.

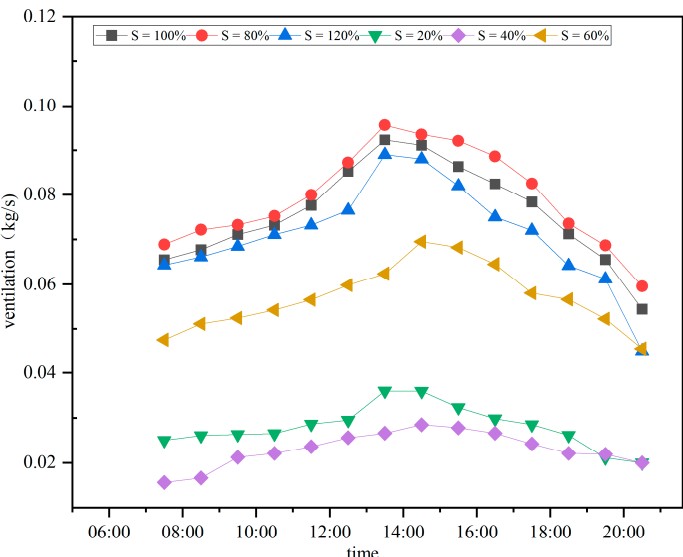

**Figure 11.** Air mass flow rates of wall-mounted solar chimneys at different times.

　　Ansys Fluent transients simulated the natural ventilation of the wall-mounted solar chimney for each moment of the day. It is very clear from Figure 11 that the overall ventilation of the wall-mounted solar chimney at S = 80% was much higher than the ventilation of the wall-mounted solar chimney at the remaining five values of S. The maximum ventilation of the wall-mounted solar chimney at S = 80% was 0.096 kg/s when the wall-mounted solar chimney received the peak amount of solar radiation. The air entering the wall-mounted solar chimney from the air inlet can be quickly preheated by solar radiation, generating a buoyancy force, thus accelerating the flow of air to the outlet and increasing the wall-mounted solar chimney ventilation.

　　With the change in S-value, the wall-mounted solar chimney ventilation gradually decreased when 20% ≤ S ≤ 60%. At 60% < S ≤ 80%, wall-mounted solar chimney ventilation showed a definite upward trend and reached a peak at S = 80%. At 80% < S ≤ 120%, wall-mounted solar chimney ventilation experienced the formation of an internal vortex, with a secondary backflow of air, resulting in a decline in ventilation. Therefore, the engineering design must focus on solving the vortex phenomenon formed by the internal airflow of the wall-mounted solar chimney.

## 5. Conclusions

　　In this simulation experiment, the effect of the ratio of the air outlet to air inlet cross-sectional area of a wall-mounted solar chimney on the operational effectiveness of a wall-mounted solar chimney system in a single-story building was investigated. Ansys Fluent, a computational fluid dynamics software, was used to simulate the airflow inside the wall-mounted solar chimney numerically. The simulation results analyzed the changes in the airflow and temperature field distribution inside the wall-mounted solar chimney at different S. The specific analysis results were as follows:

(1)　The maximum local heat flow density of the air inside the wall-mounted solar chimney reached 187.951 W/m$^2$ at S = 80%. When S > 80%, the local heat flow density inside the wall-mounted solar chimney continued to increase. It tended to flow in a steady state at S = 120%;

(2)　When S < 80%, the airflow inside the wall-mounted solar chimney did not form vortices. However, at this time, the temperature difference between the components of the air inside the wall-mounted solar chimney was slight, and the buoyancy force formed was not sufficient to provide a higher airflow rate. The natural draft of a wall-mounted solar chimney is at a low level;

(3)  At S < 80%, the wall-mounted solar chimney internal airflow state was different. When 80% ≤ S < 100%, the formation of a local vortex occurred, such that the flow of air was repeatedly heated by solar radiation. When the wall-mounted solar chimney internal air buoyancy force increased, the airflow rate increased, and wall-mounted solar chimney ventilation increased rapidly. At S = 80%, the maximum instantaneous ventilation of the wall-mounted solar chimney reached 0.096 kg/s; in the 80% ≤ S < 100% interval, the natural ventilation of the wall-mounted solar chimney was maintained at a high level;

(4)  After S ≥ 100%, the airflow in the wall-mounted solar chimney formed a vortex area which further expanded to the point of forming a secondary return flow of air. Although the flow rate of air increased, the formation of secondary reflux caused a significant loss in the efficiency of the overall ventilation of the wall-mounted solar chimney;

(5)  S-values in the design of wall-mounted solar chimneys have a non-negligible role, and the proper selection of the S-value range can make wall-mounted solar chimney ventilation achieve the actual engineering design requirements.

It is worth noting that the above results are based on the analysis of a single-story building wall-mounted solar chimney study, which had a specific scope of applicability. The current study focused on the analysis of different inlet and outlet design dimensions of wall-mounted solar energy in single-story buildings on the internal temperature and velocity fields. In future research work, the study of wall-mounted solar chimneys could be extended to multi-story buildings based on the analytical results of the S-values obtained from the current study. A combination of numerical simulations and field measurements could be used to continuously improve and expand the applicable range of S-values to improve the natural ventilation of wall-mounted solar chimneys in multi-story buildings. While optimizing the S-value, the change in air velocity inside the building due to the increase in natural ventilation of the wall-mounted solar chimney should be considered. Therefore, a comprehensive study and analysis of indoor air velocity and thermal comfort are necessary for future work.

**Author Contributions:** Methodology, Z.F. and W.W.; numerical simulation, Z.F. and Y.C.; manuscript writing, Z.F.; picture editing, Z.F. and Y.C.; Conceptualization, Z.F. and J.S. All authors have read and agreed to the published version of the manuscript.

**Funding:** This research received no external funding.

**Institutional Review Board Statement:** Not applicable.

**Informed Consent Statement:** Not applicable.

**Data Availability Statement:** The data presented in this study are available upon request from the first author.

**Conflicts of Interest:** The authors declare no conflict of interest.

## Nomenclature

| | |
|---|---|
| $A_s$ | Glass cover area ($m^2$) |
| $c_p$ | Specific heat capacity of air J/(kg·k) |
| $C_{1\varepsilon}$ | Calculation constant |
| $C_{2\varepsilon}$ | Calculation constant |
| $G_k$ | Turbulent kinetic energy due to mean velocity gradient (J) |
| $h_{1.groud}$ | Radiant heat transfer coefficient between the outside of the glass cover and the ground W/($m^2$·K) |
| $h_{1.sky}$ | Radiant heat transfer coefficient between the outer surface of the glass cover and the sky W/($m^2$·K) |
| $h_{out.1}$ | Convective heat transfer coefficient between the glass cover and the outdoor air W/($m^2$·K) |

| | |
|---|---|
| $h_{in.1}$ | Convective heat transfer coefficient between the glass cover and the air inside the chimney W/(m²·K) |
| $h_{in.2}$ | Convective heat transfer coefficient between the surface of the exterior wall and the air inside the chimney W/(m²·K) |
| $I$ | Direct solar irradiation (W/m²) |
| $I_s$ | Solar scattering irradiation (W/m²) |
| $\vec{J}$ | Diffusion flux of component |
| $k$ | Turbulent pulsation kinetic energy (J) |
| $k_{eff}$ | Effective thermal conductivity W/(m·K) |
| $M$ | Airflow rate (m/s) |
| $N_u$ | Air convection heat transfer strength between the glass cover and the cavity area W/(m²·K) |
| $p$ | Fluid pressure (Pa) |
| Pr | Prandtl number |
| $Q_{rad.wall.2}$ | Radiative heat exchange between the glass cover and the wall (W) |
| $Q_{out.air.1}$ | Convective heat exchange with air on the outside of the glass cover (W) |
| $Q_{in.air.1}$ | Convective heat exchange between the inner side of the glass cover and the inner air (W) |
| $Q_{sky.1}$ | Longwave radiation between the glass cover and the sky (W) |
| $Q_{groud.1}$ | Longwave radiation between the glass cover and the ground (W) |
| $Q_{in.air.2}$ | Convective heat exchange between the wall and the air in the runner (W) |
| $Q_w$ | The air inside the chimney absorbing heat from solar radiation through the glass cover (W) |
| $R_{in.1}$ | Heat transfer thermal resistance on the inner surface of the glass cover (m²·K)/W |
| $R_{out.2}$ | Heat transfer thermal resistance on the outer surface of the glass cover (m²·K)/W |
| Re | Reynolds number |
| $S$ | Outlet-to-inlet airflow path cross-sectional area ratio |
| $S_h$ | Heat generated by a volumetric heat source (W) |
| $T_b$ | Average temperature of the glass cover (K) |
| $T_a$ | Ambient temperature (K) |
| $T_{ground}$ | Building outdoor ground temperature (K) |
| $T_{sky}$ | Sky temperature (K) |
| $T_m$ | Average air temperature inside the chimney (K) |
| $T_{in}$ | Average indoor air temperature (K) |
| $T_u$ | Average temperature of the exterior wall surface (K) |
| $u_x$ | Velocity of the fluid in the x-direction (m/s) |
| $u_y$ | Velocity of the fluid in the y-direction (m/s) |
| $u_z$ | Velocity of the fluid in the z-direction (m/s) |
| $H$ | Height of contact between the glass cover and the internal air (m) |
| $U_\infty$ | Building outdoor airflow rate (m/s) |
| $\alpha_1$ | Absorption rate of the glass cover (0.6) |
| $\alpha_2$ | Absorption rate of the exterior wall (0.76) |
| $\alpha_k$ | Reciprocal of the effective Prandtl number of k |
| $\alpha_\varepsilon$ | Reciprocal of the effective Prandtl number of ε |
| $\beta_1$ | Transmittance of the glass cover |
| $\varepsilon_1$ | Glass cover surface emissivity |
| $\varepsilon$ | Dissipation rate of turbulent pulsating momentum |
| $\mu_{eff}$ | Fluid effective dynamic viscosity kg/(s·m) |
| $\rho$ | Air density (kg/m³) |
| $\sigma_b$ | Stefan–Boltzmann's constant $5.67 \times 10^{-8}$ W/(m²·K⁴) |

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
