# Peer review of "Structural and Heat Transfer Model Analysis of Wall-Mounted Solar Chimney Inlets and Outlets in Single-Story Buildings"

_buildings, doi:10.3390/buildings12111790_

Round 1

Reviewer 1 Report (New Reviewer)

Dear Author(s),

Your article is interesting and can meet the requirements of scientific article, however I have some questions. Bellow you can find my remarks:  .

a.       Missing citations like nbellow and add literature about different solar chimeys in for CFD since you are doing a CFD analysis, e.g : Lechowska, A.et al "CFD modelling of transient thermal performance of solar chimney used for passive ventilation in a building", IOP Conference Series: Materials Science and Engineering, 2018

b.       “In China, the energy consumption generated by building operations accounts for 20% of the total energy consumption” – add citation

c.       Through the CFD software Fluent – add citation and what version of funet you are using. The official name for flunet is Ansys Flunet.

d.       “The simulation location is still chosen as Urumqi, Xinjiang, China, and the simulation time is chosen from 07:30 to 20:30 on June 5, 2022, with an initial ambient air temperature of 28°C and maximum solar radiation of 1144.5 w/m2” – add the source of the data

Author Response

Reviewer 2 Report (New Reviewer)

The paper analyzes the effects of the inlet and outlet size of the wall-mounted solar chimney on its air flow and air temperature. the subject is of interest for the scientific community and energy efficient building industry; however, the study has structural and language issues addressed below.

Paper title

The title of the study (Structure and Energy Analysis of Wall Solar Chimney) is not responding to the content of the paper as it analyzes the effects of the inlet and outlet size on the air flow and solar chimney air temperature. The structure of the chimney was not analyzed (no comparison between different structures has been addressed) only the physical properties of the wall material is addressed.

and there is no energy analysis performed (no energy demand calculation for the different proposed scenarios) only the indirect effect on energy use is mentioned.   

Authors can consider making the title more specific

Solar wall terms

The study used different terms for the same solar chimney (Wall-mounted solar chimney, wall-type solar chimney, wall solar chimney) authors needs to define one term.

wall-mounted solar chimney could be the best

Citation

- in China, the energy consumption generated by building operations accounts for 20% of the total energy consumption. (In the first paragraph of the introduction – authors should provide a reference)

- is this part (To enhance the practical application of solar chimneys, Pandey et al. designed a solar chimney with a waste heat recovery device, using waste heat to drive the airflow inside the chimney, and the maximum conversion efficiency of the waste heat recovery solar chimney was 5.225%, which was 5% higher compared to a standard solar chimney[7].) of section 1.1 necessary? What is the relation between this research and the presented study?

- Many studies have shown that solar chimneys have excellent application prospects. Compared with rooftop solar chimneys, wall-mounted solar chimneys are more commonly used in civil buildings and are more harmoniously integrated with facilities

Authors should list those studies in the text and in the reference list

Section2.

There is no information about the height of the proposed model and the height of the solar chimney, although height is one the most important elements. Also, in this section authors needs to explain why they used only one floor?

section 3

- Authors presented many equations, which were not used in the calculation as the study used simulation tools. The authors need to explain how they used these equations and where are the results.

- Also have you established all these equations? or you are just citing them.  

- The authors mentioned (The number of grids verified under different models is shown in Table 2) (it should be table 1) as Table 2. about Solar irradiance in Urumqi at 12:00 on June 15, 2022.

section 4

Figure 4 comes before the first place it was mentioned in the text (it should come after)

section 5

In section 5.4 authors need to explain why the study used June 5, 2022 and not another date? 

Conclusion

- Authors need to expand the conclusion section to provide more details on the limitation of the study, the relation between the results from this study and previous studies, the recommendation for solar chimney design and the future work. 

Overall design.

The paper has too many sections and subsections. Authors can consider rearrangement of the paper. Here is a suggestion.

There is no need for subtitles in section 1

Section 2 and 3 can be combined as they explain the model and its properties

Consider reducing subsection in section 4

There is no 5.3 section. The section 5 includes 5.1, 5.2, and 5.4 (wrong numbering of subsections)

English language

English language is the main weakness of the paper. The paper has many long sentences, weak sentence structure and use of informal language.

Example of long sentences 

The error between the calculation results of the third set of grids and the fourth set of grids is 0.55%; when S=100%, the error between the calculation results of the third set of grids and the fourth set of grids is 0.51%; when S=120%, the error of the calculation results of the third set of grids and the fourth set of grids is 0.33%; based on the above calculation and analysis, and at the same time to ensure that the most accurate simulation results can be obtained in the limited computing resources, six models S=20%, S=40%, S=60%, S=80%, S=100%, S=120, respectively choose N4=3766896, N3=3432521, N3=3364896, N4=4064521, N4 =3341263,N4=3515478 Six sets of grids for simulation calculation. (Last paragraph of section 4.2)

Sentence structure weakness

- In the abstract the following sentence

However, in the design process of wall-mounted solar chimneys, the ventilation efficiency of wall-mounted solar chimneys is often seriously affected due to different design dimensions.

The proposed structure (However, in the design process of wall-mounted solar chimneys, the ventilation efficiency is often seriously affected due to different design dimensions.)

- In the abstract also the sentence

To better study the air flow state and airflow temperature field distribution inside the solar wall chimney, in this research work, we established six physical models of the solar wall chimney with different S (outlet to inlet air flow path cross-sectional area ratio) values.

The proposed structure (To better study the air flow state and airflow temperature field distribution inside the wall mounted solar chimney, six physical models with different S (outlet to inlet air flow path cross-sectional area ratio) values we analyzed.)

- Global warming and the frequent occurrence of extreme and severe weather remind us to save energy consumption and rationally use clean energy has become a development trend of the times; in China, the energy consumption generated by building operations accounts for 20% of the total energy consumption. (In the first paragraph of the introduction - weak structure of this sentence make it difficult to understand)

- The following sentence (first paragraph of section 1.1) also has weak structure and it’s difficult to understand

(As an extended application of passive buildings, solar chimneys use solar radiation to heat the air inside the chimney so that the internal air forms buoyancy due to temperature difference and induces indoor air flow through natural ventilation, thereby introducing fresh outdoor air and improving indoor air quality; Especially in today's disease prevention and treatment has a role that cannot be ignored)

- Also, this sentence of section 2 has weak structure (which not only improves the indoor air quality and indoor thermal environment but also saves electricity and energy, which is very important for reducing the energy consumption of building operation. This not only improves indoor air quality and indoor thermal environment but also saves electricity and energy, which is very important to reduce the energy consumption of building operations).

example of typos 

as shown in Figure 1 shown (please check the sentence)  

Authors should use the formal language and avoid the use of We

- we designed a wall-type solar chimney with an outer glass cover. We established six different S-value wall-type solar chimney models according to the ratio of the cross-sectional area of the entrance and exit of the wall-type solar chimney.

- we use CFD (Computational Fluid Dynamics) to simulate the airflow in the flow channel of the wall-type solar thermal chimney and analyze the airflow field and temperature distribution inside the solar chimney through cloud image display.

- we establish the mathematical relationship between the wall-mounted solar chimney's air mass flow rate and the S value 

Author Response

Reviewer 3 Report (New Reviewer)

File attached.

Author Response

Reviewer 4 Report (New Reviewer)

The manuscript concentrated on the effect of various outlet/inlet cross-sectional area ratios (S) of wall-mounted solar chimneys on the performance of natural ventilation in buildings. The temperature and velocity field analysis is conducted by Fluent software for six different S values, and the results are discussed. The novelty of the study is described well, but the construction and presentation of the manuscript should be improved. 

My suggestions that would improve the quality of the manuscript:

1. Introduction section should start with a description of solar chimneys, types of solar chimneys, specifications and application areas. Please include figures. 

2. The type of solar chimney evaluated in the manuscript is the wall-mounted solar chimney. In the text, different terms are used such as wall solar chimney, wall-type solar chimney. Please use the same term throughout the text. 

3. Some sentences are repeated in the text such as Section 1.2, Paragraph 1, and Section 2. Please check the whole manuscript.

4. Definition of S value should be presented better. 

5. The methodology should include diagrams to explain Eq. 1-5. Heat inputs/outputs and temperatures used in the equations should be visible in the diagrams. The same applies to Sections 3.2 and 3.3.

6. Abbreviations should be given in parentheses not another way around.

7. Table 1 is mentioned in the text as Table 2. 

8. The font of Table 1 is different than the rest of the text.

9. The last paragraph of the Results section includes the aim of the manuscript. This part should be included in the Introduction section. Performance parameters mentioned here and the definition of "conversion efficiency of the chimney" which was mentioned in the introduction should be explained.

10. Could you please discuss the air flowrate and velocity requirements of the indoor spaces for thermal comfort and environmental air quality? Therefore, you need to define a criterion to discuss the results of the air flowrate. 

11. Since nomenclature exists, variables of the equations do not need to be explained just below the equations. Or remove the nomenclature. 

12. If the nomenclature remains, please check the units and definitions. I noticed an example: M is air flowrate, not air quality flowrate.

13. Conclusions should be included future work direction. 

Author Response

Reviewer 5 Report (New Reviewer)

Overall, the topic is interesting and worthy of investigation. However, I do not think that this current paper should be considered for publication owing to many issues concerning with it. First, no critical analysis of the literature has been provided which makes it hard to see the contribution of this study. There are also some issues with the research methodology as highlighted below. Lastly, I believe that no implications have been provided on the findings of this study. Hence, I have to, unfortunately, reject your paper. Below, I have some comments that I hope would be useful for authors to improve the paper.

1.       Introduction

This section has been poorly written, please revise the entire section. Also, please add a reference to back up the statements you are making, e.g., “in China, the energy consumption generated by building operations accounts for 20% of the total energy consumption.”

Reading through the introduction section, I failed to see any elaboration given on the gap of knowledge and the need for doing this current study. Yes, there have been randomly cited studies in line 6 of section 1.2 but no critical analysis has been provided on the shortages/lacks associated with these studies. I have seen no clear research questions to be addressed. To me, this paper can be rejected at this stage.

What is the relevance of sections 2 and 3 here? What do these sections aim to elaborate?

4. Simulation method

I found this section very weak! For instance, why SpaceClaim has been chosen? What are the features of this software that make it suitable for doing these analyses? Plus, we usually develop a section to acknowledge the limitations concerned with the study, when the research approach is simulation. This is also missing.

Lastly, the results are simply presented without elaborating on the implications of their findings. 

Round 2

Reviewer 2 Report (New Reviewer)

The authors have responded to most of the comments

Authors can consider reducing the sub-sections of section 3 (for example: setting and boundary conditions can be in one sub-section)  

In the conclusion section authors could address the limitation of the study

Author Response

Reviewer 3 Report (New Reviewer)

Dear Authors, than you for so complete answer, and thank you for the cooperation :-)

P.S. Please correct "w/m²" (still occurs at least 9 times in the text). The correct unit is “W/m²” (capital letter "W" is required for watts).

Author Response

Reviewer 4 Report (New Reviewer)

The authors mainly improved the manuscript based on the comments, but there are still some missing ones.

1.       Repetition still exist. An example in the Introduction:

"The many studies mentioned above show that the research on the effect of external influences on the ventilation of solar chimneys is well established. The aforementioned literature focuses on the influence of external factors on the operational effectiveness of solar chimneys".

The authors should check the manuscript once again. 

2. One of my comments was:  "Definition of S value should be presented better". The response to my comment was, "Based on your suggestions, we have redefined the S-value air outlet to inlet cross-sectional area ratio (S)". 

But this definition was given in the Abstract only. Please do not use abbreviations in the abstract and give the definition first in the Introduction. 

3. My other comment was, "If the nomenclature remains, please check the units and definitions. I noticed an example: M is air flowrate, not air quality flowrate". The authors replied as they corrected this, but it is not.

4. Furthermore, the nomenclature should be in alphabetic order, and greek letters should be given as another list.

5. Please use "multi-storey building" instead of "multi-story building"

6. In my 10th comment, I asked to discuss the air flowrate and velocity requirements of the indoor spaces for thermal comfort and environmental air quality?. The authors replied as this will be future work. But I can not see this recommendation in the conclusions. 

7. Even though it was improved, the manuscript still needs to be revised based on grammar and technical language. As an example, the following sentence has no verb. 

"In this simulation study, in order to analyze the effect of the ratio of air outlet to air inlet cross-sectional area of a wall-mounted solar chimney on the operational effectiveness of a wall-mounted solar chimney system in a single-story building"

Author Response

Reviewer 5 Report (New Reviewer)

The authors have addressed my comments appropriately, hence I recommend this paper for publication.

Author Response

This manuscript is a resubmission of an earlier submission. The following is a list of the peer review reports and author responses from that submission.

Round 1

Reviewer 1 Report

The article is interesting, the theme is relevant. Some considerations must be done in order to improve the quality of the manuscript.

1. Abstract. Please consider to explain better the problem before announcing what is the aim of this research, presented in the article. Also important to describe brieflyin the abstract what is the Method. 

2. Introduction, section 1.1. In the second paragrafh the word "altitude" seems to refer to the location (city) of the experiment and not to the height of the chimney. Please find another word.
In the section 1.2. some sentences are repeated "External factors...such as ambiente temperature, ambiente wind speed, intensity of solar radiation and positition of building windows". Please try another way to explain these two cases with similar external factors without so many repetitions of the same factors/elements.

3. Item 2, Model description. First paragrah the word "shown" is repeated.

4. The figure 1, Model drawing of wall solar chimney: the first figure seems to be a Section, but this is not explained. The numbers and letters are not readable. I suggest to enlarge figures and numbers. The second image is not very useful at al, it doesn´t shows any important aspect, please consider to eliminate it.  About the figures, it would be interesting to have one illustration to show how the chimney works in a multistorey building. This is not clear and this use is mentioned on the description.

5. About Method, as the study used a computational simulation of CFD, a better description of the available softwares would be important. It´s important to relate all the formulae presented in 4.2. with the software used. Is it possible to program the software and insert the equations on it? Please explain

6. On 4. Simulation Method, the values of S values are presented in percentage. Please refer to the values mentioned before (0,6, 0,8, 1,0). Uniformize it. On 4.3, Solver Setting, please explain where is the location of the study (Urump): country, latitude, longitude and so on.

7. In Results and Discussion, figures 2 and 3 are not cited in the text. Also figure 5 is not cited. Figure 6 is mentioned, but instead of the figure an equation is presented. Please organize this.
Generally figures and graphs must be cited and explored in the text.

Reviewer 2 Report

The most important specific remarks.

1. The symbol list (Nomenclature) is incomplete; many of the symbols used are not explained either in the nomenclature or in the manuscript text.

2. The units of some physical quantities are misspelled.

3. It is not clear what the relationship is between Egs. (1) - (14) and (15) - (20).

4. Why are equations (15) - (20) written for the transient fluid flow and heat transfer conditions while the steady state is being analyzed?

5. The description of the CFD model is absolutely imprecise, incl. no information about the applied numerical grid, grid-independent test and many other important details concerning numerical model applied is given.

6. The presentation of the results and the discussion are badly organized, incl. Fig. 3 is not mentioned in the text, Fig. 5 is discussed before Fig. 4, fig. 8 is discussed first, and Fig. 7 thereafter.

7. The physical meaning of the natural ventilation volume M (equation (21)) is unclear.

8. The use of the term "optimal values" for S = 0.8 and S = 1 is unjustified.

9. The graphical presentation of the results shown in Figs. 3 and 5 is strange - the rainbow type (color) of the bars is unjustified.

10. It should be clearly stated that the results presented in Fig. 9 were obtained theoretically. The calculation results (CFD) should be verified experimentally. The validation of the CFD model is necessary to make the calculation results more reliable.

11. In the Nomenclature, some (non-dimensionless) physical quantities do not have units.

12. No pages are provided for many of the articles listed in the References.